# Advances and Challenges in SARS-CoV-2 Detection: A Review of Molecular and Serological Technologies

**DOI:** 10.3390/diagnostics14050519

**Published:** 2024-02-29

**Authors:** Mai M. El-Daly

**Affiliations:** 1Special Infectious Agents Unit-BSL3, King Fahd Medical Research Center, King Abdulaziz University, Jeddah 21589, Saudi Arabia; meldaly@kau.edu.sa; Tel.: +966-566736711; 2Department of Medical Laboratory Sciences, Faculty of Applied Medical Sciences, King Abdulaziz University, Jeddah 21589, Saudi Arabia

**Keywords:** SARS-CoV-2, COVID-19, molecular detection methods, serological techniques

## Abstract

The urgent need for accurate COVID-19 diagnostics has led to the development of various SARS-CoV-2 detection technologies. Real-time reverse transcriptase polymerase chain reaction (RT-qPCR) remains a reliable viral gene detection technique, while other molecular methods, including nucleic acid amplification techniques (NAATs) and isothermal amplification techniques, provide diverse and effective approaches. Serological assays, detecting antibodies in response to viral infection, are crucial for disease surveillance. Saliva-based immunoassays show promise for surveillance purposes. The efficiency of SARS-CoV-2 antibody detection varies, with IgM indicating recent exposure and IgG offering prolonged detectability. Various rapid tests, including lateral-flow immunoassays, present opportunities for quick diagnosis, but their clinical significance requires validation through further studies. Challenges include variations in specificity and sensitivity among testing platforms and evolving assay sensitivities over time. SARS-CoV-2 antigens, particularly the N and S proteins, play a crucial role in diagnostic methods. Innovative approaches, such as nanozyme-based assays and specific nucleotide aptamers, offer enhanced sensitivity and flexibility. In conclusion, ongoing advancements in SARS-CoV-2 detection methods contribute to the global effort in combating the COVID-19 pandemic.

## 1. Introduction

Coronavirus disease 2019 (COVID-19), caused by infection with severe acute respiratory syndrome coronavirus 2 (SARS-CoV-2), presents a range of symptoms in humans, varying from mild to critical. The global spread of this pandemic has been rapid since its inception in China in December 2019. Despite extensive containment efforts to control the disease, the virus has persisted with varying clinical manifestations in numerous countries. A coordinated approach that includes the precise diagnosis and understanding of the disease spread, monitoring, and preventive measures is crucial to manage the crisis effectively. Accurate and rapid diagnostic technologies play a pivotal role [1].

A valid diagnostic test is the most effective method to identify cases within a large population, including those without symptoms, trace the pathways of transmission and individuals carrying the virus, as well as evaluate therapeutic efficacy, and confirm infection eradication. Therefore, investing in state-of-the-art technologies and offering financial support to create and validate reliable tests for COVID-19 has become a top priority for each country. These tests are crucial tools for tracing, isolating, and treating outbreaks [2].

This review provides an overview of various nucleic acid-based and serological techniques currently accessible for COVID-19 diagnosis, aiming to assist physicians and clinical microbiologists in selecting appropriate methods for the diagnosis and clinical management of COVID-19.


**SARS-CoV-2 Genome and Structure**


Coronaviruses belong to the Coronaviridae family within the Nidovirales order. They are characterized as enveloped, positive single-stranded RNA viruses [3]. Four genera are recognized: delta (δ), gamma (γ), alpha (α), and beta (β) [4]. Phylogenetic analysis reveals a close relationship between SARS-CoV-2 and β-coronaviruses [1,5]. Like other coronaviruses, SARS-CoV-2 has a 5’-cap and a 3’-UTR poly(A) tail in its positive-sense single-stranded RNA genome. The genome is less than 30 kb in size and has 14 open reading frames (ORFs) that code for structural proteins such as spike (S), envelope (E), membrane/matrix (M), and nucleocapsid (N), as well as accessory proteins and non-structural proteins (NSPs) that are essential for virus replication, as shown in Figure 1 [6,7]. The initial ORF constitutes almost 65% of the viral genome, which translates to either polyprotein pp1a (nsp1–11) or pp1ab (nsp1–16). NSP3, NSP9, NSP10, NSP12, NSP15, and NSP16 are the six NSPs that are essential for viral replication; structural and accessory proteins are encoded by the remaining ORFs [8,9].

The spike (S) protein is a transmembrane structure comprising receptor binding (S1) and cell membrane fusion (S2) subunits that help viral envelopes bind to angiotensin-converting enzyme 2 (ACE2) receptors on the surfaces of host cells [10]. The nucleocapsid (N) protein binds to the viral genome and is essential for immune evasion, virion production, and RNA replication. It interacts with nsp3 and M proteins for these functions [10]. The matrix (M) protein, characterized by high abundance and conservation, plays a key role in the assembly and budding of viral particles. This is accomplished by interactions with accessory proteins 3a and 7a, as well as N [11,12]. The smallest protein in SARS-CoV-2, the envelope (E) protein, aids in the production, development, and release of virions [8].

Growing evidence indicates that the SARS-CoV-2 genome has experienced evolutionary changes and diversification as it has spread geographically. The comprehensive genomic analysis of SARS-CoV-2 isolates from around the world has revealed numerous genomic regions characterized by increased genetic variation and unique mutation patterns [1], as shown in Figure 2.

According to a recent study by Korber et al. (2020), it was determined that a particular amino acid alteration, known as D614G, in the virus spike protein, emerged in the early stages of the pandemic. Currently, viruses carrying G614 are widespread globally. The researchers proposed a hypothesis suggesting that the rapid dissemination of G614 may be linked to its heightened infectiousness compared to D614. To substantiate this hypothesis, the authors presented evidence showing that clinical samples from G614 infections demonstrate increased levels of viral RNA and generate higher titres in pseudo viruses during in vitro experiments [13].


**Distribution of SARS-CoV-2 in Body Fluids and Tissues**


SARS-CoV-2 undergoes a transition from the upper respiratory to the lower respiratory tract, where it undergoes replication and dissemination via respiratory aerosol. Consequently, the confirmation of viral infection is achieved through the collection of nasal or oropharyngeal samples, either independently or in combination [14]. Early in the illness, the virus load in respiratory samples is most noticeable. It peaks in the second week of the illness and then starts to decline. In cases of severe illness, the concentration of the virus in respiratory fluids is highest during the third and fourth weeks. Interestingly, viral RT-qPCR assays performed on throat swabs from patients who have recovered from the disease can produce positive results for up to 50 days [15].

The virus can be identified in samples collected from tracheal aspirates, bronchoalveolar lavage, pleural fluids, urine, blood, and faeces [15]. Notably, anal, and faecal swabs have shown viral RNA even weeks after negative results from respiratory samples [16]. Saliva is also considered a viable substitute supply for SARS-CoV-2, as well as virus-specific antibodies. It may emerge as a standard for extended journeys, such as air or sea travel, to ensure that individuals are not infected with the SARS-CoV-2 virus. An infection is indicated by a positive salivary viral antigen test [17,18,19].

Notably, stool samples might harbour the virus for as long as four weeks after the onset of the disease. The well-recognized risk of exposure of healthcare workers to the faeces of an infected person becomes particularly significant during procedures that generate a high level of aerosols. The European Centre for Disease Prevention and Control (ECDC) has advised continuing self-isolation because of the prolonged period of virus shedding seen in both faeces and respiratory samples up to 14 days following discharge [14].

The detection of SARS-CoV-2 in faeces has recently raised concerns about a faecal-oral or faecal-respiratory transmission pathway. As a result, more exact detection of the virus in faeces is critical for understanding and controlling viral spread. However, faecal samples are far more complicated than nasopharyngeal/oropharyngeal swabs, sputum, and other respiratory samples, so it is critical to develop an appropriate methodology to implement optimal sampling and RNA extraction procedures, as biases introduced by this process may influence detection. The SARS-CoV-2 positivity rate in faecal samples was associated with the severity of sickness. Cao and colleagues used four different real-time RT-PCR kits to detect SARS-CoV-2 in faeces samples (DAAN, Sansure, Bio-Germ, and GeneoDx) [20].

Lastly, the detection of SARS-CoV-2 in wastewater provides a method for community monitoring and proves to be a powerful tool for tracking the transmission of COVID-19. Screening tests for sewage are now available for dormitories, allowing the detection of asymptomatic individuals. In the case of a positive outcome, institutions can implement quarantine measures for the infected individuals, thus preventing potential outbreaks of SARS-CoV-2 [14].


**Collection of COVID-19 Samples**


Antigens and virions are detectable in specimens obtained from nasopharyngeal swabs or saliva [21]. However, before delving into detection methods, addressing pre-analytical considerations in the collection and storage of nasopharyngeal swabs is crucial. The handling of nasopharyngeal swabs in the pre-analytical stage significantly influences the reliability of results. Pondaven-Letourmy et al. reported an approximate 30% false-negative rate for RT-qPCR, despite its high sensitivity, emphasizing the importance of proper nasopharyngeal sampling [22,23,24,25]. Considering that SARS-CoV-2 relies on the spike protein’s binding to the ACE2 cellular receptor for cell infection, it is noteworthy that ACE2 expression is higher in the distal part of the nose than in the proximal part [26]. Assuming standardized nasopharyngeal swab collection techniques, attention must shift to storage conditions. Basso et al. investigated the impact of different conditions of storage on the sensitivity of the RT-qPCR test.

Optimal results, indicated by the lowest threshold cycles, were obtained when nasopharyngeal swabs were kept at +4 °C in an extraction buffer solution for RNA preservation. Additionally, the researchers found that it is reliable to store swabs in viral transport media at room temperature for up to two days prior to RT-qPCR testing; however, cooling is advised for longer periods of time [27].

Nasopharyngeal swabs should be kept in sterile, dry, sealed plastic tubes for antigen testing purposes. Swabs stored in viral transport media should have a capacity of no more than one millilitre and should not include guanidinium. Swabs can be kept in the refrigerator for one day at +2–8 °C or at room temperature for up to eight hours [28]. To ensure the precision and dependability of COVID-19 diagnostic results, these factors are crucial.


**SARS-CoV-2 Detection Methods**


The fast spread of COVID-19 has underscored the immediate requirement for precise diagnostic technologies. Currently, the available diagnostic tools rely on (a) molecular methods for viral gene detection, (b) the detection of human antibodies, and (c) methods for detecting viral antigens. Among these techniques, the detection of viral genes by RT-qPCR has been identified as the most reliable method, as outlined in Figure 3 [29].


**A. Viral Gene Detection by Molecular Methods**


Molecular methods include the following. (1) Nucleic acid amplification techniques (NAATs) constitute the dominant methodologies utilized in research and clinical environments. These include real-time reverse transcriptase polymerase chain reaction (RT-qPCR), sequencing, and digital PCR (dPCR). (2) Isothermal amplification techniques include loop mediated isothermal amplification (LAMP), nucleic acid sequence based amplification (NASBA), recombinase polymerase amplification (RPA), clustered regularly interspaced short palindromic repeats (CRISPR)-related amplification, and others. Each technique possesses distinct and unique characteristics, allowing for optimal performance concerning sensitivity, specificity, stability, simplicity, and cost-effectiveness. The unmet demand for NAATs is particularly notable in molecular diagnostics, especially in point-of-care (POC) scenarios [30].

## 2. Nucleic Acid Amplification Techniques (NAATs)

### 2.1. Real-Time Reverse Transcriptase Polymerase Chain Reaction (RT-qPCR)

The diagnosis of COVID-19 has employed RT-qPCR since the beginning of the outbreak [31]. This process entails the collection of nasal or oropharyngeal swabs from individuals, wherein reverse transcriptase converts extracted RNA into complementary DNA (cDNA). Subsequently, a targeted sequence is amplified using DNA polymerase and specific primers [32,33]. To enhance the reliability of the assay, it is advisable to amplify two or more genes, including one specific and one conserved gene. This has led to the development of various RT-qPCR tests employing diverse gene combinations [34,35].

During replication, the SARS-CoV-2 viral genome undergoes ongoing changes, resulting in the formation of fresh waves of pandemic episodes. Early obtained virions, namely the reference genome (SARS-CoV-2, NC_045512.2), are used to construct PCR primers [5,36]. Consequently, even a single mutation within a primer sequence can compromise the performance of RT-qPCR assays, potentially resulting in false negative detection outcomes [37]. Analyses of sequencing data submitted to GenBank and GISAID show those mutations are primarily found in the ORF1ab region, particularly in Germany and China [38,39].

Notably, all the targets of the COVID-19 diagnostic primers recommended by the US CDC had mutations. In contrast, many mutations in various clusters were found using N gene primers and probes utilized in China, Japan, and Thailand. This suggests that RT-qPCR kits may not find the N gene to be a stable target, highlighting the necessity to regularly upgrade N gene-based assays to account for novel variations like alpha, beta, gamma, and delta [40].

Although RT-qPCR is considered the gold standard for detecting SARS-CoV-2 due to its high sensitivity and specificity, it does have some disadvantages: (1) RT-qPCR assays are time-consuming, typically requiring several hours to complete, including RNA extraction, reverse transcription, PCR amplification, and result interpretation, leading to delays in obtaining test results, especially in high-demand settings. (2) RT-qPCR is complex and involves multiple steps and requires specialized laboratory equipment and trained personnel. (3) Despite its high sensitivity, RT-qPCR can yield false negative results, particularly when the viral load in the sample is low or the sample collection is not performed properly. (4) RT-qPCR testing can be relatively expensive, especially when considering the costs of laboratory equipment, reagents, and personnel. (5) RT-qPCR detects active infection only through viral RNA, indicating active viral infection at the testing time. Despite these disadvantages, RT-qPCR remains a critical tool for SARS-CoV-2 detection, particularly in clinical diagnosis and surveillance. Efforts to address these limitations include the development of rapid POC assays and the integration of RT-PCR with other diagnostic methods.

### 2.2. Sequencing for Diagnostic Purposes

Viral genome sequencing is not ideal for quick detection in large populations due to its greater costs, increased requirements for data interpretation, and inferior clinical efficiency when compared to RT-qPCR. However, a metagenomic RNA sequencing technique was able to successfully retrieve the original genome sequence of SARS-CoV-2 [5]. According to reports from the WHO and China, 104 SARS-CoV-2 strains were identified and sequenced using Illumina and Nanopore technologies between late December 2019 and mid-February 2020 [41]. The GISAID and GenBank databases have over 1000 sequences that are identical to SARS-CoV-2, allowing for a full understanding of the genomic and proteomic components of the virus [42].

The advantage of sequencing-based detection is the ability to track viral mutations by gathering data on new strains. This approach aids in the identification and classification of evolving coronavirus strains [41]. The careful tracking of viral evolution data shows that as the virus replicates and spreads, random alterations in the genome accumulate at a rate of roughly two per month [43]. There have been reports of notable novel mutant viruses, such as delta (B.1.617.2), gamma (P.1), beta (B.1.351), and alpha (B.1.1.7), which may increase the danger of accelerated virus transmission [44,45]. High-throughput techniques and portable rapid sequencing technologies have been developed as COVID-19 diagnostic tools in response to growing demand [2].

### 2.3. Digital PCR

An advancement in technology over the popular RT-qPCR techniques is digital PCR (dPCR). It is further divided into droplet digital PCR (ddPCR) and chip digital PCR (cdPCR). ddPCR works by micro-partitioning samples and achieving DNA ultra-dilutions via a water–oil emulsion in the reaction mix. Taq polymerase is used in a conventional PCR reaction using ddPCR to amplify a target DNA fragment from the sample using a pre-validated primer or probe assays. Before amplification, in the ddPCR process, the reaction is divided into thousands of distinct reactions prior to amplification. Furthermore, in ddPCR, data are collected at the reaction endpoint [28,46]. Applications of digital PCR include low-copy-number genes, microbiological studies, and the identification of lowly expressed targets [47,48].

ddPCR outperforms other molecular approaches in terms of repeatability, stability, sensitivity, and specificity [49]. It enables physicians to diagnose COVID-19 infection in individuals who are asymptomatic or paucisymptomatic by accurately detecting extremely low viral loads [50]. According to a recent study, 63 samples that had previously tested negative for SARS-CoV-2 by RT-qPCR were positive for the virus using ddPCR, and 55% of those samples displayed COVID-19 symptoms [51]. Its optimal limits of detection (LODs) for the N and SARS-CoV-2 ORF1ab genes are 1.8 copies/reaction and 2.1 copies/reaction, respectively, demonstrating its remarkable precision. In experimental circumstances, RT-qPCR revealed a LOD of 1039 copies/reaction for ORF1ab and 873.2 copies/reaction for the N gene. Compared to RT-qPCR, ddPCR is 500 times more sensitive [52].

Moreover, the sensitivity of molecular testing was increased by ddPCR’s high sensitivity in identifying the SARS-CoV-2 N gene in individuals who had been mistakenly classified as negative by RT-qPCR [53]. It has been shown that the detection rate can be increased by using ddPCR to identify viral loads in sputum samples as opposed to nasal swabs [45]. Because of this, ddPCR’s higher sensitivity beats RT-qPCR in the clinical diagnosis of COVID-19, which lowers the frequency of false negative results [47].

## 3. Isothermal Amplification Techniques

While RT-qPCR stands as a reliable and sensitive assay, its drawback lies in its time-consuming nature, requiring approximately 2 h for the assay itself and an additional 60 min for RNA extraction [54]. In contrast, isothermal amplification assays operate at a consistent temperature throughout the reaction, as these techniques amplify a target sequence at a single temperature. As a result, a thermocycler is no longer needed to sustain the required temperature changes during the typical stages of the RT-PCR process [55]. Some methods of isothermal amplification include loop-mediated isothermal amplification (LAMP), nucleic acid sequence-based amplification (NASBA), recombinase polymerase amplification (RPA), and clustered regularly interspaced short palindromic repeats (CRISPR). Each of these strategies has significant advantages in terms of simplicity and efficiency, contributing to isothermal amplification’s versatility [46].

### 3.1. Reverse Transcriptase Loop-Mediated Isothermal Amplification (RT-LAMP)

Loop-mediated isothermal amplification (LAMP), a widely adopted amplification method developed by Notomi et al. in 2000 [56], has gained popularity in recent times. When applied to SARS-CoV-2 amplification, LAMP necessitates modifications, incorporating Bst DNA polymerase, nucleotides, reverse transcriptase, and a specific set of primers, two outer primers, two inner primers, and two loop primers. This modified method, known as RT-LAMP, operates at a temperature range of 60–65 °C and amplifies the target sequence in just 1 h with high specificity, similar to standard RT-PCR. Remarkably, LAMP can generate 10^9^ DNA copies within this short timeframe [56].

Considering that the SARS-CoV-2 virus’s nucleic acid is RNA, reverse transcription is used to generate cDNA for RT-LAMP. Following that, the LAMP primers bind to their complementary target sequence in the cDNA, resulting in the formation of dumbbell-shaped DNAs. The Bst DNA polymerase, with its intrinsic strand displacement activity, amplifies the dumbbell-shaped DNA, yielding products of various sizes by the end of the reaction. The results obtained from the RT-LAMP reaction can be interpreted using turbidity, colorimetric assays, or fluorescence measurements [32,57,58].

Amaral et al. found that RT-LAMP demonstrates 100% sensitivity and 96.1% specificity in detecting fewer than 100 viral genome copies of SARS-CoV-2 in just 30 min. Additionally, they introduced a brand new colorimetric detection technique based on the interaction of divalent zinc (Zn^2+^) with the complex metric indicator murexide (MX). In the presence of Zn^2+^, murexide is yellow, and when Zn^2+^ is absent, it is pink [59].

Notably, because isothermal amplification occurs at approximately 62–65 °C, RT-LAMP eliminates the need for expensive temperature change instruments. This feature distinguishes LAMP as a method that has the potential to be adapted for use in portable devices. An incubation period of 30–60 min is required for the detection of SARS-CoV-2 using the RT-LAMP method [28]. Reverse transcription and LAMP can be combined in a single reaction due to the versatility of RT-LAMP.

RT-LAMP has several advantages for the amplification of nucleic acids, such as reduced time for amplification, resistance to inhibitory substances in clinical specimens, and compatibility with a broad variety of commercially accessible reagents. This technique is employed for RNA template amplification, exhibiting strong strand displacement activity and temperature endurance, in addition to being quicker and requiring fewer resources [60]. Later developments, including extending two-loop primers, have significantly lowered the time for initial LAMP reaction [61].

The versatility of RT-LAMP extends beyond SARS-CoV-2, with applications in pathogen identification for MERS-CoV, West Nile virus, Zika virus, yellow fever virus, Ebola virus, and various other pathogens [62,63,64,65,66,67,68]. The effectiveness of RT-LAMP in identifying SARS-CoV-2 was assessed in a recent study by El-Kafrawy et al. in 2022. The researchers improved the RT-LAMP technique by contrasting two fluorescence amplification mixtures and various reaction periods. Subsequently, the outcomes were contrasted with those acquired using conventional real-time RT-PCR. To perform RT-LAMP on samples diluted 1:4 in water treated with diethylpyrocarbonate (DEPC), several sets of singleplex and multiplex LAMP primers targeting the N, S, and ORF1ab genes were used. The mixture was incubated at 65 °C with a real-time PCR 7500. Remarkably, this methodology showed 100% agreement with the industry standard protocol for SARS-CoV-2 nucleic acid detection. The researchers suggest that the test is a good fit for POC detection in public hospitals and medical centres [69].

### 3.2. Nucleic Acid Sequence-Based Amplification (NASBA)

This approach requires fewer cycles than RT-qPCR and LAMP and can produce a large copy number in 1.5–2.0 h using a two-stage isothermal amplification experiment. This technique’s stability, sensitivity, and specificity in detecting single-stranded RNA (ssRNA) make it highly promising for POC applications, especially in low-resource situations [70]. NASBA was successfully used to detect SARS-CoV during previous epidemics, with real-time monitoring made possible using fluorochromes in the reaction. A modified version called Multiplex RT-NASBA could simultaneously identify several viral infections. Xing et al. [71] analysed 614 clinical samples from patients with respiratory tract infections using the RTisochipTM-W system. Of them, 201 were preclinical, 25 were clinically diagnosed, and 14 were COVID-19 positive. In a single run, this system can analyse sixteen samples in ninety minutes.

### 3.3. Recombinase Polymerase Amplification (RPA)

Results are obtained in 10 to 15 min using an isothermal approach that amplifies DNA sequences at 37 to 42 °C. In this diagnostic method, the recombinase enzyme first interacts with primers to generate a recombinase–primer complex. The primer looks for its complementary sequence and resides in the target site while the recombinase breaks down the target strand. A single-stranded DNA-binding protein stabilises the other strand while DNA polymerase lengthens the broken strand. Future rounds of extension can still use the recombinase [72].

RT-RPA is a more advanced version of RPA designed for coronavirus detection that includes a reverse transcription step. Lau et al. created an RT-RPA procedure in which the endpoint can be determined in 15–20 min using SYBR green or a lateral flow strip technique. According to their results, RT-RPA has a 100% specificity and 98% sensitivity for detecting 7.659 C/μL [73]. Similar RT-RPA protocols were built by other groups as well, such as Wang et al., 2021 [74].

A method for quick diagnosis and POC testing called microfluidic integrated lateral flow recombinase polymerase (MI-IF-RPA) was created by Liu et al. Extracted RNA was loaded into the chip’s inlet holes with the RT-RPA reaction buffer and primers specific to the SARS-CoV-2 N gene. Afterwards, the chip was incubated at 42 °C for 15 min. The results were analysed using the lateral flow assay, where two bands indicate a positive sample and one band indicates a negative sample. The chip displayed 100% specificity and 97% sensitivity, and it exhibited no cross-reactivity with other viruses such as coronavirus OC43, respiratory syncytial, or influenza. This suggests that an RT-RPA assay based on chips could be created [75].

### 3.4. CRISPR-Based Detection Technique

The method known as clustered regularly interspaced short palindromic repeats (CRISPR) may identify any type of nucleic acid present in a sample, including RNA and DNA. It has several advantages compared to conventional diagnostic methods. This system offers a fast turnaround time, excellent sensitivity and specificity, and an affordable and dependable diagnostic platform. Additionally, by removing the requirement for thermocycler applications, CRISPR facilitates POC testing [76,77].

Viral load is often used in traditional approaches to assess findings; low viral load individuals are regarded as negative. In contrast, the CRISPR technique relies on identifying the viral genome, which minimizes the possibility of false negative results and permits early infection identification. Furthermore, this technology can distinguish between mutant strains within the population because it is intended to produce guide RNA (gRNA) unique to a conserved area of the virus genome.

Broughton et al. created the SARS-CoV-2 DNA endonuclease-targeted CRISPR trans reporter (DETECTR), a CRISPR-based detection method. The procedure begins with RNA extraction from the collected sample, followed by reverse transcription to produce cDNA. Isothermal amplification techniques are then used to amplify the cDNA, with primers designed to assess the amplification of the SARS-CoV-2, N, and E genes. A lateral flow assay, facilitated by a FAM-biotinylated reporter molecule, can yield results in 30–40 min [78].

Hou et al. compared metagenomic sequencing and RT-PCR to a CRISPR-based diagnostic method. They created CRISPR-COVID, a fast detection test that uses RT-RPA to amplify the SARS-CoV-2 ORF1a gene. Notably, this method takes only 40 min to complete, which is a significant improvement over the 20–24 hours required for RT-PCR analysis and metagenomic sequencing. The fluorescence readout determines the assay results, which reveal a SARS-CoV-2 specificity of 100% [79].

There are various methods to detect SARS-CoV-2, including gold nanoparticle-based biosensors. These biosensors have been widely used in different scientific fields including medical care, biology, chemistry, and physics. They serve as signal amplifiers and resonance light scattering for viral detection. Moitra and colleagues [80] have developed a technique that can detect COVID-19 in just 10 min. The approach is relatively simple, and it involves the use of plasmonic gold nanoparticles that change colour in the presence of the virus, hence making the test positive. This method detects viral RNA on the first day of infection, and the RNA can be extracted from the sample within 10 min. The test uses small gold particles to detect specific proteins. When coupled with the virus gene sequence, the gold nanoparticles change the colour of the liquid reagent from purple to blue. This test’s accuracy is determined by its ability to detect the virus, which many commercially available test methods cannot achieve until several days after infection. The authors claim that this test’s development and use are significantly cheaper than laboratory tests since it does not require laboratory equipment or qualified personnel to administer and examine [80].


**B. Serological assays**


In disease surveillance, detecting antibodies against a virus in infected patients is an important diagnostic approach. Even though RT-qPCR is the most widely used technology for detecting active SARS-CoV-2 infections, viral RNA becomes practically undetectable 14 days after sickness, and false negative results are possible due to inappropriate viral sample handling. These challenges underline the importance of developing simple test kits based on the detection of human antibodies generated in response to viral infection. The main principle behind antibody-based immunodiagnosis is to identify antibodies (IgG and IgM) produced in response to viral infection and/or viral antigen using an enzyme-linked immunosorbent assay (ELISA) [34].

According to research, antigen-specific antibodies can be discovered in a patient after 3 to 6 days, with IgG being detectable later in an infection [1]. These tests are excellent for use in resource-constrained labs since they can offer information on both ongoing and historical infections. They can also be used in disease surveillance programs to improve the understanding of the infection rate in the community. While serological assays can detect both ongoing and past infections, they are most efficient in verifying SARS-CoV-2-specific antibody responses to detect past infections [1].


**Detection of SARS-CoV-2 antibodies**


The production of antibodies against SARS-CoV-2 is the first line of defence against infection. By day 7, neutralizing antibodies can be found in as many as 50% of infected people, and by day 14, all infected people have them. For SARS-CoV-2 diagnostics, serological investigations provide an alternative to RT-qPCR, and the combination of serological testing and real-time PCR greatly increases the rates of positive virus identification. IgG antibodies suggest previous SARS-CoV-2 exposure, whereas IgM antibodies indicate recent viral infection. As a result, immunodiagnostic assays are essential for assisting in the creation of COVID-19 vaccines and for determining the level of infection in people who are not actively infected [81].

In most people, IgM levels rise during the first week after SARS-CoV-2 infection, peak after two weeks, and subsequently decline to near-normal levels, owing to the significant demand for quick tests in identifying COVID-19 infections [82]. After a week, IgG is visible and stays raised for a long time—up to 48 days at times—possibly offering protection against reinfection. After infection, IgA responses usually start to manifest 4–10 days later. In addition to IgG and IgM, serum IgA is used as a diagnostic predictor [83,84]. Differential target antigens have an impact on the range of SARS-CoV-2 antibodies. Antibody titres may drop seven days following an infection [82].

According to current research, antibodies specific to SARS-CoV-2 have been discovered in saliva [81]. To evaluate the variations in antibody levels between sera and saliva, multiplex SARS-CoV-2 antibody immunoassays were used. Parallel compartmental humoral immune responses are suggested by antibodies in saliva that match those in sera [81]. Salivary IgA is correlated with the severity of COVID-19 disease, and a fast immunoassay was developed in parallel using the BreviTest platform technology to measure salivary IgA [73].

It is interesting to note that low IgA levels can suggest herd immunity even in cases when no viral exposure has been documented [85]. Antibodies specific to SARS-CoV-2, especially those found in saliva, may be detected for surveillance purposes. Still up for debate, though, are the best antigens to use in serological testing. Although the viral S protein is a good option, it is still unclear which precise area of the S protein should be targeted. As an alternative, different S protein isoforms, such as those seen in different strains, could be used to guarantee the repeatability of the assay [86].

Testing time might range from 13 min (Abbott) to 45 min (Cepheid Xpert Xpress) to receive results [87]. Two colloidal gold immunoassays (VivaDiag COVID-19 IgG-IgM based and Assay Genie rapid POC kit), two lateral-flow immunoassays (SureScreen rapid test cassette and BioMedomics rapid test), and one time-resolved fluorescence immunoassay (Goldsite diagnostics kit) are available.

A clinical study will be required to assess clinical significance. SARS-CoV-2 IgG (Abbott) sensitivity in N-based immunoassays can reach 100%. In the case of S-based immunoassays, the Liaison SARS-CoV-2 S1/S2 IgG and the combined S- and N-based platform COVID-19 VIRCLIA IgG MONOTEST showed comparable sensitivity. The neutralization test for plaque reduction had a sensitivity of 93.3%. To assess specificity, all assays, except for the enzyme-linked immunosorbent assay (IgG) (EUROIMMUN), produced at least one positive result for the negative SARS-CoV-2 antigen control. This most likely suggests that assay sensitivity has changed significantly over time and between testing platforms [88].

Despite being the gold standard for immunoglobulin-based detection, the plaque reduction neutralization test has some limitations, including a restricted number of sample analyses and the requirement of a biosafety level 3 laboratory. There is a strong association between the plaque reduction neutralization test and the titres produced by these assays. Antibody tests are currently used mostly for epidemiological testing [14,88].


**C. SARS-CoV-2 Antigens**


The N and S proteins are the primary immunogens in SARS-CoV [89]. One recent fast diagnostic test that used novel antigens demonstrated increased sensitivity and specificity in patients who were symptomatic and had a high viral load [90]. In contrast, a quick method that used a fluorescent immune chromatographic assay to find the N protein demonstrated maximum sensitivity only in the early stages of infection [91]. According to a mass spectrometry study, gargle solution samples from COVID-19 patients included the N protein [92].

For almost 80 years, mass spectrometry has been recognized as one of the most sensitive procedures in the literature. Although a genetic probe can detect the presence of virus RNA in a biological sample, mass spectrometry can confirm the presence of the generated protein, and immunofluorescence can then emphasize the iconographic aspects of the contacts. Spectral counting is a semi-quantitative evaluation in terms of absent or low protein and/or peptide presence [93]. In vitro, bacterial cultures can be evaluated for increased or decreased protein concentrations during infection, as well as the expansion of viral particle peptide numbers. Compared to statistical analysis and immunofluorescence microscopy in samples, this method allows data to be cross-referenced, with each serving as a control for the other. Mass spectrometry can detect the presence of D-amino acids in peptides. As a result, the immunofluorescence microscopy technique can be integrated with the spectral counting of viral peptides in bacterial cultures using mass spectrometry, as well as identifying D-amino acids within viral peptides in bacterial cultures and patient blood. Previous studies [94,95] have described the genetic aspect of viral replication in bacterial cultures, interactions by electron microscopy and immunofluorescence by light microscopy, evidence of the nitrogen isotope N15 introduced into 30-day bacterial cultures, and presence in viral proteins after another seven days, all of which serve as the foundation for detailed analysis of genetic–molecular aspects [96].

According to Petrillo et al.’s study, the use of immunofluorescence and spectral counting in mass spectrometry seems to aid in identifying viral proteins in tested samples. This method also enables the detection of various viral proteins produced during bacterial growth in an in vitro investigation. It is critical to highlight the bacteriophage behaviour of SARS-CoV-2, as well as other viruses. Increasing signal intensity by fluorescence microscopy with fluorescent antibodies in fractions of bacterial cultures containing the virus under a study conducted for up to 30 days appears to be increasingly crucial for ignoring potential bacteriophage behaviour. New insights into the presence of D-amino acid in SARS-CoV-2 proteins and toxin-like peptides detected in the plasma of COVID-19 patients could help us understand the virology mechanism of RNA viruses that can bind and interact with epithelial cells as well as bacteria in the human gut microbiome [94].

Furthermore, 73.6% of COVID-19 patients with confirmed cases had their urine samples test positive for the N protein using a fluorescence immunological chromatographic assay [91]. The S protein is thought to be more suited for detection during the recovery phase since it manifests later in the infection [97]. A microplate reader makes it easy to perform an ultrasensitive antigen test tailored for the S protein [98].

In addition to established methods for detecting the SARS-CoV-2 coronavirus nucleocapsid antigen, a half-strip lateral flow (HSLF) assay has been developed. This assay has higher clinical sensitivity than standard serology assays, with an LOD of 3.03 ng/mL for the commercially available Genscript N protein [99]. Another novel strategy uses a nanozyme-based chemiluminescent paper assay that can be used with common cell phones and has a LOD of 0.1 ng/mL for the SARS-CoV-2 recombinant spike antigen [100]. An antibody’s specificity in identifying the target is matched when using a particular nucleotide aptamer against the N protein for antigen detection. However, it offers the potential for heightened sensitivity and greater flexibility in developing assays for various purposes [2].

## 4. Conclusions

In conclusion, the ongoing developments in SARS-CoV-2 detection methods address the critical need for accurate and efficient COVID-19 diagnostics. Molecular techniques, including nucleic acid amplification techniques (NAATs) and isothermal amplification techniques, offer reliable viral gene detection. Serological assays, particularly those detecting antibodies in saliva, play a crucial role in disease surveillance and contribute to our understanding of both active and past infections. While various rapid tests, including lateral-flow immunoassays, show promise for quick diagnosis, their clinical significance requires validation through further studies. Challenges such as variations in specificity and sensitivity among testing platforms and evolving assay sensitivities over time need careful consideration. SARS-CoV-2 antigens, notably the N and S proteins, are central to diagnostic methods, and innovative approaches like nanozyme-based assays and specific nucleotide aptamers offer enhanced sensitivity and flexibility. All things considered, the ongoing progress in SARS-CoV-2 detection techniques is a major asset to the international response to the COVID-19 pandemic, providing several valuable instruments for prompt and precise diagnosis.

## Figures and Tables

**Figure 1 diagnostics-14-00519-f001:**
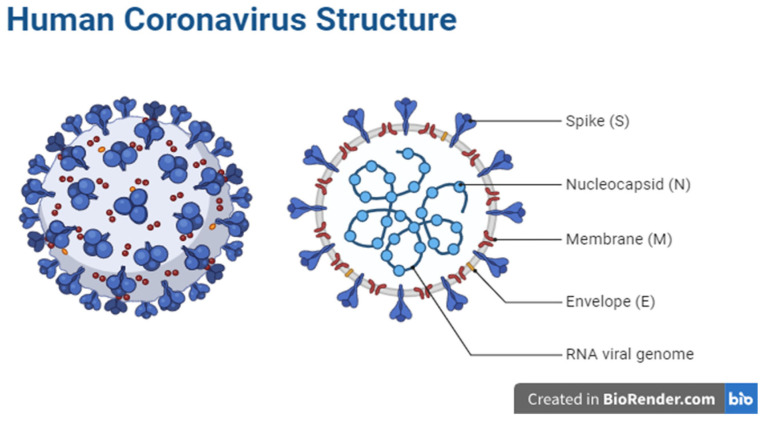
Illustrates the structure of the human coronavirus. The virus has different components that work together to infect host cells and reproduce. These components include the spike (S) protein which binds to receptors on the surfaces of host cells, enabling the virus to enter the cells. The nucleocapsid (N) protein binds to the viral RNA genome, forming the nucleocapsid. The membrane (M) protein helps shape the viral envelope and is involved in viral assembly. The envelope (E) protein plays a role in the assembly and release of the virus from host cells. The RNA genome contains the genetic material of the virus.

**Figure 2 diagnostics-14-00519-f002:**
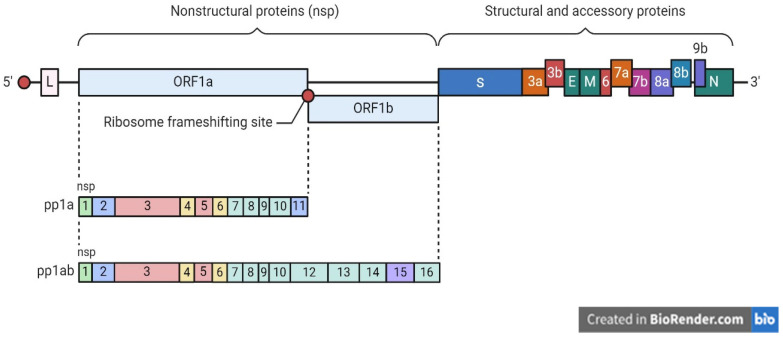
Illustrates the genome of SARS-CoV-2, which is composed of several components, including the 5’ and 3’ UTRs. The genome contains genes responsible for structural proteins, including the spike (S) protein that attaches to receptors on host cells, the envelope (E) protein that aids in the release of the virus from host cells, the membrane (M) protein that helps shape the viral envelope and is involved in viral assembly, and the nucleocapsid (N) protein that binds to the viral RNA genome, forming the nucleocapsid. Additionally, it encodes non-structural proteins that contain the open reading frames (ORFs); these are the coding regions of the genome that translate into viral proteins.

**Figure 3 diagnostics-14-00519-f003:**
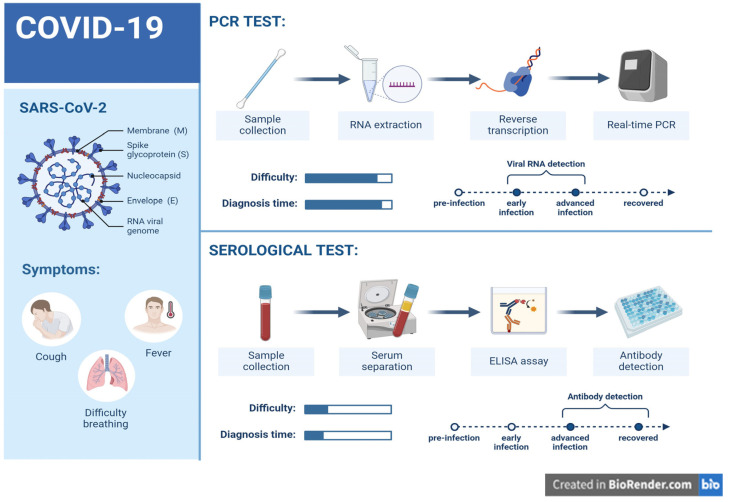
Illustrates various methods used to diagnose and monitor SARS-CoV-2 infections. These methods include the following: (1) Reverse transcription real-time PCR is a variation of PCR used to amplify viral RNA by first converting it to DNA. It is the most common way to detect SARS-CoV-2. (2) Serological tests are a type of diagnostic test that can detect the presence of antibodies in the blood. These antibodies can indicate whether a person has been infected with the SARS-CoV-2 virus in the past.

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
