# Peer review of "Advances and Challenges in SARS-CoV-2 Detection: A Review of Molecular and Serological Technologies"

_diagnostics, 2024, doi:10.3390/diagnostics14050519_

Round 1
Reviewer 1 Report
Comments and Suggestions for Authors
This review article summarized the diagnostic methods for the detection of SARS-CoV-2 infection, including molecular methods, serological technologies, and rapid antigen tests.
Several suggestions:
1. Title, [SARS-CoV-2] not [SAR-CoV-2].
2. Based on [threshold cycles] in line 134 and [RT-qPCR] in line 177, the most commonly mentioned method is [Real-time Reverse Transcriptase Polymerase Chain Reaction (Real-time RT-PCR) or quantitative Reverse Transcriptase Polymerase Chain Reaction (RT-qPCR). Please use (Real-time RT-PCR) or (RT-qPCR) not RT-PCR in this article, which represents Reverse Transcriptase-Polymerase Chain Reaction.
3. Line 16, is there a [other]] before [molecular methods]?
4. In Fig. 3, why use [salivary samples] not other samples, such as NPS? No [cellular and soluble mediator analysis] is mentioned in this article. No [in vitro infectivity] is mentioned in this article.
5. Fig. 2 is not directly related to diagnosis. It is better to show the [Schematic diagrams of the SARS-CoV-2 genome] and mention where is the locations to define the various variants [alpha, beta, delta, micron in line 186] or the locations for molecular detection.
6. In the entire article, [ORF1ab] or [orf1ab]?
7. Line 254, reference 55 is not for SARS-CoV-2 detection, please check it.
8. Lines 291, 319, 334, please use [point-of-care (POC)] at its first appearance, then use [POC]. The same applied to [Limit of Detection (LOD)].
9. Section from lines 227 to 233, [ddPCR] or [cdPCR]?
10. Please add references after lines 245, 363, 365, and 380, [had a sensitivity of 93.3%] in line 412.
11. Line 423-424, why add [and antibodies against these proteins can remain in SARS patients' blood for up to 30 weeks] here, it is misleading.
12. It is better to mention the disadvantages of other molecular methods than RT-qPCR if available.
Author Response
Reviewer 1
Several suggestions:
Thank you to the reviewer for providing valuable comments.
- Title, [SARS-CoV-2] not [SAR-CoV-2].
I appreciate the reviewer's comment, thank you. My apologies for this error.
I realized that there was a typing mistake in my previous text and have now corrected it.
“Advances and Challenges in SARS-CoV-2 Detection: A Review of Molecular and Serological Technologies”
- Based on [threshold cycles] in line 134 and [RT-qPCR] in line 177, the most commonly mentioned method is [Real-time Reverse Transcriptase Polymerase Chain Reaction (Real-time RT-PCR) or quantitative Reverse Transcriptase Polymerase Chain Reaction (RT-qPCR). Please use (Real-time RT-PCR) or (RT-qPCR) not RT-PCR in this article, which represents Reverse Transcriptase-Polymerase Chain Reaction.
Thank you for the reviewer’s comment.
All instances of "real-time Reverse Transcriptase Polymerase Chain Reaction" were changed to "RT-qPCR" throughout the text.
- Line 16, is there a [other] before [molecular methods]?
Thank you for the reviewer’s comment.
The word “other” added to the sentence. “While other molecular methods”
- In Fig. 3, why use [salivary samples] not other samples, such as NPS? No [cellular and soluble mediator analysis] is mentioned in this article. No [in vitro infectivity] is mentioned in this article.
Thank you for the reviewer’s comment.
The old figure was removed, and a new one was added in its place.
- 2 is not directly related to diagnosis. It is better to show the [Schematic diagrams of the SARS-CoV-2 genome] and mention where is the locations to define the various variants [alpha, beta, delta, micron in line 186] or the locations for molecular detection.
Thank you for the reviewer’s comment.
The original figure was removed, and a new one was added in a different location to depict genome organization.
- In the entire article, [ORF1ab] or [orf1ab]?
Thank you for the reviewer’s comment.
The correct term is ORF1ab, and it has not been changed.
“Several sets of singleplex and multiplex LAMP primers targeting the N, S, and ORF1ab genes were used”.
- Line 254, reference 55 is not for SARS-CoV-2 detection, please check it.
Thank you for the reviewer’s comment.
Reference document 55 details the application of the loop-mediated isothermal amplification (LAMP) method for general DNA amplification and detection.
- Lines 291, 319, 334, please use [point-of-care (POC)] at its first appearance, then use [POC]. The same applied to [Limit of Detection (LOD)].
Thank you for the reviewer’s comment.
I have made corrections to all the abbreviations throughout the entire text.
- Section from lines 227 to 233, [ddPCR] or [cdPCR]?
Thank you for the reviewer’s comment.
It's ddPCR.
- Please add references after lines 245, 363, 365, and 380, [had a sensitivity of 93.3%] in line 412.
Thank you for the reviewer’s comment.
The following references have been added in the order mentioned: [46], [34], [1], [81], and [88].
All added references are highlighted.
- Line 423-424, why add [and antibodies against these proteins can remain in SARS patients' blood for up to 30 weeks] here, it is misleading.
Thank you for the reviewer’s comment.
The sentence has been deleted.
- It is better to mention the disadvantages of other molecular methods than RT-qPCR if available.
Although RT-qPCR is considered the gold standard for detecting SARS-CoV-2 due to its high sensitivity and specificity, it does have some disadvantages: (1) Time-consuming, RT-qPCR assays typically require several hours to complete, including RNA extraction, reverse transcription, PCR amplification, and result interpretation, leading to delays in obtaining test results, especially in high-demand settings. (2) Complexity, RT-qPCR involves multiple steps and requires specialized laboratory equipment and trained personnel. (3) False negatives: despite its high sensitivity, RT-qPCR can yield false-negative results, particularly when the viral load in the sample is low or the sample collection is not done properly. (4) Cost: RT-qPCR testing can be relatively expensive, especially when considering the costs of laboratory equipment, reagents, and personnel. (5) Detection of active infection only: RT-qPCR detects viral RNA, indicating active viral infection at the testing time. Despite these disadvantages, RT-qPCR remains a critical tool for SARS-CoV-2 detection, particularly in clinical diagnosis and surveillance. Efforts to address these limitations include the development of rapid, POC assays and the integration of RT-PCR with other diagnostic methods.
Page 6, lines 203-217
Reviewer 2 Report
Comments and Suggestions for Authors
The review of Mai M. El-Daly entitled Advances and Challenges in SAR-CoV-2 Detection: A Comprehensive Review of Molecular and Serological Technologies describes a correct and full evaluation of molecular methods of SARS-CoV-2 protein diagnosis. the review describes all methods and their importance.
There are no form errors and the literature is appropriate. The manuscript requires a small intervention in the extension of an additional method and an additional topic covering the virus. Methods of finding in fecal samples and the importance of proteomics (mass spectrometry) should also be described. There are several studies on this. In addition, the importance of the bacteriophage mechanism of the virus and the importance of checking this aspect as well during viral research to anticipate mutations should also be described.
Author Response
Thank you to the reviewer for providing valuable comments.
The following paragraphs have been added to the manuscript.
The detection of SARS-CoV-2 in faeces has recently raised concerns about a fecal-oral or fecal-respiratory transmission pathway. As a result, more exact detection of the virus in faeces is critical for understanding and controlling viral spread. However, faecal samples are far more complicated than nasopharyngeal/oropharyngeal swabs, sputum, and other respiratory samples, so it is critical to develop an appropriate methodology to implement optimal sampling and RNA extraction procedures, as biases introduced by this process may influence detection. The SARS-CoV-2 positivity rate in faecal samples was associated with the severity of sickness. Cao and colleagues used four different real-time RT-PCR kits to detect SARS-CoV-2 in faeces samples (DAAN, Sansure, Bio-Germ, and GeneoDx) [20].
Page 3-4, lines 115-124.
There are various methods to detect SARS-CoV-2, including gold nanoparticle-based biosensors. These biosensors have been widely used in different scientific fields including medical care, biology, chemistry, and physics. They serve as signal amplifiers and resonance light scattering for viral detection. Moitra and colleagues [80] have developed a technique that can detect COVID-19 in just 10 minutes. The approach is relatively simple, and it involves the use of plasmonic gold nanoparticles that change color in the presence of the virus, hence making the test positive. This method detects viral RNA on the first day of infection, and the RNA can be extracted from the sample within 10 minutes. The test uses small gold particles to detect specific proteins. When coupled with the virus's gene sequence, the gold nanoparticles change the color of the liquid reagent from purple to blue. This test's accuracy is determined by its ability to detect the virus, which many commercially available test methods cannot do until several days after infection. The authors claim that this test's development and use are significantly cheaper than laboratory tests since it does not require laboratory equipment or qualified personnel to administer and examine [80].
Page 10, lines 386-400.
For almost 80 years, mass spectrometry has been recognized as one of the most sensitive procedures in literature. Although a genetic probe can detect the presence of virus RNA in a biological sample, mass spectrometry can confirm the presence of the generated protein, and immunofluorescence can then emphasize the iconographic aspects of the contacts. Spectral counting is a semi-quantitative evaluation in terms of absent or low protein and/or peptide presence [93]. In vitro bacterial cultures can be evaluated for increased or decreased protein concentrations during infection, as well as the expansion of viral particle peptide numbers. Compared to statistical analysis and immunofluorescence microscopy in samples, this method allows data to be cross-referenced, with each serving as a control for the other. Mass spectrometry can detect the presence of D-amino acids in peptides. As a result, the immunofluorescence microscopy technique can be integrated with spectral counting of viral peptides in bacterial cultures using mass spectrometry, as well as identifying D-amino acids within viral peptides in bacterial cultures and patient blood. Previous studies [94, 95] have described the genetic aspect of viral replication in bacterial cultures, interactions by electron microscopy and immunofluorescence by light microscopy, evidence of the nitrogen isotope N15 introduced into 30-day bacterial cultures, and presence in viral proteins after another seven days, all of which serve as the foundation for detailed analysis of genetic-molecular aspects [96] .
According to Petrillo et al.'s study, the use of immunofluorescence and spectral counting at mass spectrometry seems to aid in identifying viral proteins in tested samples. This method also enables the detection of various viral proteins produced during bacterial growth in an in vitro investigation. It is critical to highlight the bacteriophage behavior of SARS-CoV-2, as well as other viruses. Increasing signal intensity by fluorescence microscopy with fluorescent antibodies in fractions of bacterial cultures containing the virus under a study conducted for up to 30 days appears to be increasingly crucial for ignoring potential bacteriophage behavior. New insights into the presence of D-Amino acid in SARS-CoV-2 proteins and toxin-like peptides detected in the plasma of COVID-19 patients could help us understand the virology mechanism of RNA viruses that can bind and interact with epithelial cells as well as bacteria in the human gut microbiome [94].
Page 12, lines 482-510
Round 2
Reviewer 1 Report
Comments and Suggestions for Authors
This revised manuscript has addressed the issues I raised previously. Please re-write the figure legends, especially figures 1 and 3.
Author Response
Figure 1 legend:
Figure 1 illustrates the structure of the Human Corona Virus. The virus has different components that work together to infect host cells and reproduce. These components include the Spike (S) protein which binds to receptors on the surface of host cells, enabling the virus to enter the cells. The Nucleocapsid (N) protein binds to the viral RNA genome, forming the nucleocapsid. The Membrane (M) protein helps shape the viral envelope and is involved in viral assembly. The Envelope (E) protein plays a role in the virus's assembly and release from host cells. The RNA genome contains the virus's genetic material.
Figure 2 Legend:
The genome of SARS-CoV-2 is composed of several components, including the 5' and 3' UTRs. The Open Reading Frames (ORFs) are the coding regions of the genome that are translated into viral proteins. Among the viral proteins are the Spike (S) protein which binds to receptors on the surface of host cells. The Envelope (E) protein plays a role in the assembly and release of the virus from host cells. The Membrane (M) protein helps shape the viral envelope and is involved in viral assembly. The Nucleocapsid (N) protein binds to the viral RNA genome, forming the nucleocapsid.
Figure 3 legend
Figure 3 illustrates various methods used to diagnose and monitor SARS-CoV-2 infections. These methods include: (1) Reverse Transcription real-time PCR: This is a variation of PCR used to amplify viral RNA by first converting it to DNA. It is the most common way to detect SARS-CoV-2. (2) Serological tests are a type of diagnostic test that can detect the presence of antibodies in the blood. These antibodies can indicate whether a person has been infected with the SARS-CoV-2 virus in the past.